

# A random forest algorithm for the prediction of cloud liquid water content from combined CloudSat/CALIPSO observations

Richard M. Schulte[1], Matthew D. Lebsock[2], John M. Haynes[3], and Yongxiang Hu[4]

[1]Department of Atmospheric Science, Colorado State University, Fort Collins, CO, USA
[2]NASA Jet Propulsion Laboratory, California Institute of Technology, Pasadena, CA, USA
[3]Cooperative Institute for Research in the Atmosphere, Fort Collins, CO, USA
[4]NASA Langley Research Center, Hampton, VA, USA

*Correspondence to*: Richard M. Schulte (rick.schulte@colostate.edu)

**Abstract.** A significant fraction of liquid clouds are not captured in existing CloudSat radar-based products because the clouds are masked by surface clutter or have insufficient reflectivities. To account for these missing clouds, we train a random forest regression model to predict cloud optical depth and cloud top effective radius from other CloudSat and CALIPSO observables that do not include the radar reflectivity profile. By assuming a subadiabatic cloud model, we are then able to retrieve a vertical profile of cloud microphysical properties for all liquid-phase oceanic clouds that are detected by CALIPSO's lidar but missed
by CloudSat's radar. Daytime estimates of cloud optical depth, cloud top effective radius, and cloud liquid water path are robustly correlated with coincident estimates from the MODIS instrument onboard the Aqua satellite. This new algorithm offers a promising path forward for estimating the water contents of thin liquid clouds observed by CloudSat and CALIPSO at night, when MODIS observations that rely upon reflected sunlight are not available.

## 1 Introduction

Low-level liquid clouds play a vital role in Earth's climate system, influencing radiative balance (e.g., Hartmann et al., 1992) and weather patterns (e.g., Ma et al., 1996). These clouds cool the climate by reflecting incoming solar radiation, and changes in the extent, thickness, or properties of these clouds in the future could have important implications. Indeed, the low cloud feedback is one of the most important sources of uncertainty in global climate models (Zelinka et al., 2016). Satellite datasets of low clouds can provide near-global coverage using consistent instruments, and thus are well suited for evaluating and
constraining cloud models. While many different instruments can be used to estimate low-cloud fraction, the *CloudSat* satellite (Stephens et al., 2008), with its 94-GHz Cloud Profiling Radar (CPR; Tanelli et al., 2008) is particularly noteworthy because of its ability to provide vertically resolved estimates of cloud liquid water content (LWC). These vertical profiles can be used for process studies, model validation, and to calculate shortwave and longwave radiative heating profiles (Henderson et al. 2013).



The CloudSat Data Processing Center (DPC) currently produces two operational retrievals of cloud water content. The first, 2B-CWC-RO, is a "radar-only" product that relies only on profiles of reflectivity from CPR and is based upon optimal estimation (Austin et al., 2009). The second, 2B-CWC-RVOD (Leinonen et al., 2016), is a daytime-only product that is further constrained by visible wavelength optical depth measurements from the Moderate Resolution Imaging Spectoradiometer (MODIS; Justice et al., 1998) onboard the *Aqua* satellite, which flew in formation with CloudSat as part of

NASA's "A-Train" of satellites from 2006-2018. These products have proven to be quite valuable to the scientific community (e.g., Yue et al., 2020; Ham et al., 2022; Oreopoulos et al., 2022). However, both 2B-CWC-RO and 2B-CWC-RVOD only provide estimates of cloud water for radar bins that are deemed "likely cloud" by the CloudSat cloud mask algorithm (Marchand et al., 2008). In practice, this means that the cloud must return CPR reflectivities that are above the radar's noise floor, which was around -30 dBZ at the beginning of the mission (Tanelli et al., 2008), and have a cloud top high enough so

as not to be masked by surface clutter. As a result, many low-altitude, shallow liquid clouds are not captured in the operational cloud water content products (Christensen et al., 2013; Li et al., 2018; Lamer et al., 2020; Schulte et al., 2023). This is particularly problematic for radiation studies, as even relatively thin liquid clouds can reflect substantial incoming solar radiation (Turner et al., 2007).

    Another member of the A-Train is the *CALIPSO* satellite (Winker et al., 2009), which carries the Cloud-Aerosol Lidar

with Orthogonal Polarization (CALIOP; Hunt et al., 2009). CALIOP can detect the presence and cloud-top phase of even very thin clouds, although it lacks the cloud profiling capabilities of CPR because its signal rapidly attenuates in liquid clouds. When comparing the percentage of time CALIOP detects a single layer low-level (below 5 km) liquid cloud to the percentage of the time these clouds are detected by CloudSat, it becomes clear that the operational CloudSat products fail to capture many of the clouds detected by CALIOP. Figure 1 shows the 2009 daytime single-layer warm cloud fraction from 2B-CWC-RO and

2B-CWC-RVOD compared to MODIS and CALIOP. In some of the stratocumulus dominated areas of the world, CALIOP detects a liquid cloud in close to 80% of CloudSat pixels, while the CPR cloud fraction is less than half that. MODIS cloud fractions are also not quite as high as those from CALIOP, indicating that it too misses many of the thinnest clouds, but they are still much higher than the cloud fractions from CPR.

    Schulte et al. (2023) demonstrated a method of estimating profiles of cloud water using MODIS measurements of

cloud optical depth ($\tau$) and cloud top effective radius ($r_e$). With this method, the cloud top height is determined by CALIOP, and the vertical distribution of the cloud water is calculated using adiabatic parcel theory, modified to account for the fact that observed clouds are often subadiabatic (Wood et al., 2009). This method produces reasonable estimates of cloud liquid water contents for clouds which are detected by CALIOP and MODIS but not by CPR. However, as demonstrated in Fig. 1, some thin liquid clouds are seen by CALIOP but missed even by MODIS. Moreover, MODIS observations rely upon reflected

sunlight, so this method is not viable at night.

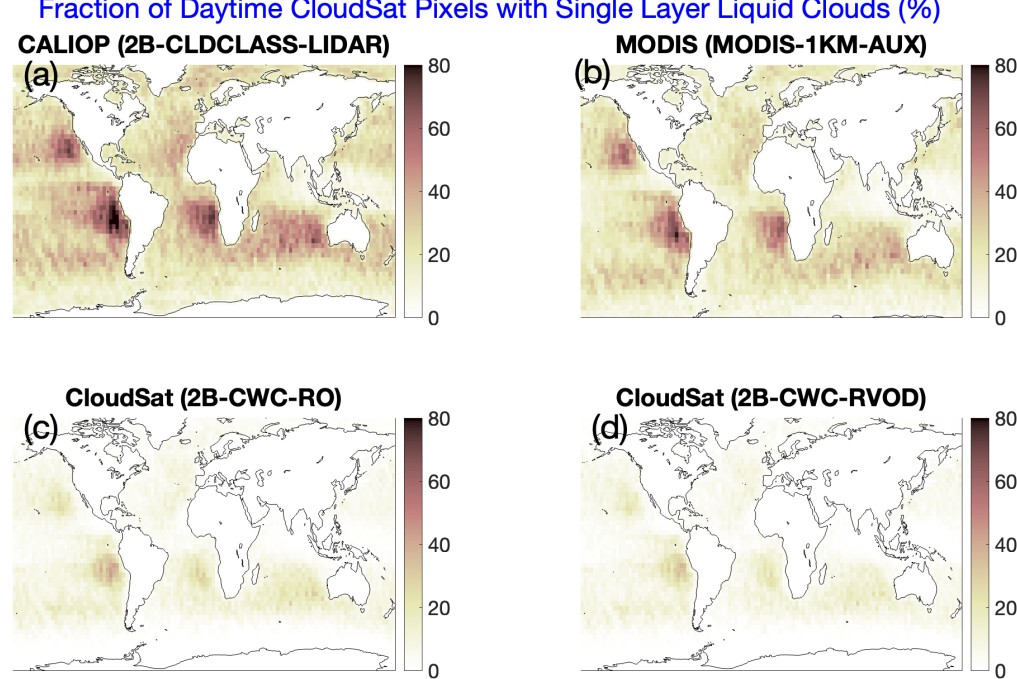

**1Figure 1: Percentage of daytime 2009 CloudSat oceanic pixels that contain a single layer liquid cloud below 5 km, according to various R05 CloudSat products: (a) 2B-CLDCLASS-LIDAR, (b) MODIS-1KM-AUX, (c) 2B-CWC-RO, and (d) 2B-CWC-RVOD.**

Motivated by these limitations of the MODIS subadiabatic model, in this study we develop a random forest machine
learning model to predict $\tau$ and $r_e$ from *non-radar-reflectivity* observables from CPR and CALIOP. Then the same subadiabatic assumptions can be used to produce a profile of LWC for clouds that are detected by CALIOP but do not have associated CPR reflectivities or MODIS cloud microphysical retrievals. The methodology is detailed in Section 2, the model performance evaluated in Section 3, and in Section 4 we offer our conclusions.

**2 Data and methods**

In this study we make use of several operational data products obtained from the CloudSat DPC (https://www.cloudsat.cira.colostate.edu). In all cases, we use the R05 version of each product. CPR geolocation data and the surface backscatter cross section come from 2B-GEOPROF (Marchand et al., 2008), while the 94 GHz brightness temperature ($TB_{94}$), derived from the radar noise floor in non-cloudy radar bins and available for all CloudSat pixels, is found in the 2B-TB94 product (Lebsock and Suzuki, 2016). Auxiliary atmospheric information comes from ECMWF-AUX, including total
column water vapor (TCWV), sea surface temperature (SST), 10 m wind speed, and profiles of temperature and pressure (used by the subadiabatic model). This data is from the European Centre for Medium-Range Weather Forecasts (ECMWF) HRES





(high resolution) forecast model, collocated to the CPR profiles by the DPC. We use the 2B-CLDCLASS-LIDAR product (Sassen et al., 2008) to screen for clouds detected by CALIOP, determine the phase of the clouds, and set the cloud top height. For training our model, we use MODIS 3.7 $\mu$m channel estimates of $\tau$ and $r_e$ from MODIS-1KM-AUX. These data are

provided at 1 km resolution; we use the 1 km MODIS pixel whose center is closest to the center of the CPR footprint for each matchup. Finally, we compare our estimates of cloud water content to estimates from 2B-CWC-RVOD and 2B-CWC-RO.

We obtain additional CALIOP data as follows. 532 nm column integrated attenuated backscatter (CIAB) and column optical depth derived from the Ocean Derived Column Optical Depths algorithm (ODCOD $\tau$) come from the CALIOP Level 2 1 km Cloud Layer Product, version 4.51 (CAL_LID_L2_01kmCLay-Standard-V4.51). We also use CALIOP-derived

estimates of cloud top effective radius (CTER) and cloud top LWC (CTLWC). These are generated using the methodology of Hu et al. (2021) and can be found on the DPC website. Similar to MODIS, these CALIOP products are provided at 1 km resolution, and we follow the same colocation procedure as for MODIS data.

**2.1 Screening**

For this study we use A-train data from 2008 and 2009. These years are chosen because there are fewer interruptions in data

availability than in many other years, and because CPR and CALIOP were still "young" and functioning at their highest capabilities. Only CloudSat granules that have valid output files for all of the R05 products mentioned above are included. In addition, our focus is only on oceanic CloudSat pixels which contain single layer liquid phase clouds. To screen for this, we require that 2B-CLDCLASS-LIDAR indicate that a given pixel have only one cloud layer, that the layer is liquid, and that the top of the layer is at or below 5 km above sea level. In addition, the 2B-GEOPROF land/sea flag must be equal to "2" (indicating

ocean). 2008 is used to train the random forest model, while 2009 is used to evaluate the retrieval performance. After this screening is applied, we are left with 24,645,411pixels from 2008 and 21,643,449 pixels from 2009. These amount to 15.7 % and 15.8 % of all CloudSat observations in 2008 and 2009, respectively. It is worth noting that the ENSO index was negative for the entirety of 2008 (La Niña conditions), while 2009 began with a negative ENSO index but was positive by the end of the year. Thus, the climate state was slightly different during the test period compared to the training period.

Before the ODCOD data is ingested into our algorithm, there is a small amount of pre-processing that is done in addition to the colocation procedure described above. The ODCOD column $\tau$ estimates include a bit-wise quality flag. If bit 10 indicates that no lidar surface was found, we consider the column optical depth signal to be saturated and we arbitrarily set the ODCOD $\tau$ value to 5. The point of this is to distinguish pixels for which there is no ODCOD estimate because the column is too optically thick for CALIOP to see the ocean surface from pixels for which there is no ODCOD estimate for other reasons.

The value of 5 is chosen because it is larger than all other ODCOD $\tau$ estimates for which the signal is not saturated. In theory, it should not matter which value of $\tau$ is chosen to represent saturated pixels, as the random forest method makes no assumptions of linearity in the input-output relationships. In practice, we tested setting the value of saturated pixels to either 50 or 500 instead of 5 and in both cases the effect on retrieval performance was minimal.



## 2.2 Sub-adiabatic cloud model

The concept of using cloud optical depth and droplet effective radius to infer cloud water content has been around for decades (e.g., Stephens, 1978). However, to do so, one must make assumptions about the vertical structure of the cloud. Two common approaches are to either assume that the cloud is vertically homogeneous (e.g., Nakajima and King, 1990), or to assume an "adiabatic cloud," one in which the cloud water linearly increases from base to top, while droplet number concentration stays constant (e.g., Brenguier et al., 2003). Both assumptions lead to closed-form expressions for the integrated LWP of a cloud as

a function of $\tau$ and cloud top $r_e$ (Wood and Hartmann, 2006). However, liquid clouds in the real world do not fit neatly into either of these two categories (Brenguier et al., 2000; Rangno and Hobbs, 2005; Rauber et al., 2007; Min et al., 2012).

Schulte et al. (2023) used an adjustment to the adiabatic model (Wood et al., 2009) meant to account for cloud processes such as entrainment and mixing that tend to cause actual clouds to have subadiabatic growth rates. With this model, the LWC $l$ of a cloud increases with height $h$ above cloud base according to Eq. (1):

$$l(h) = c(T,P)\, h\,\frac{z_0}{z_0+h},\tag{1}$$

where $z_0$ is a scaling factor and $c(T,P)$ is given by Eq. (2):

$$c(T,P) = \rho_{air}\,\frac{c_p}{L_v}(\Gamma_d - \Gamma_m).\tag{2}$$

$c(T,P)$ is the moist adiabatic condensation rate at temperature $T$ and pressure $P$, with $\rho_{air}$ equal to the air density of a fully saturated air parcel at that temperature and pressure. $c_p = 1004$ J/kg K is the specific heat of dry air at constant pressure, $L_v=$

$2.26 \times 10^6$ J/K is latent heat of vaporization of water, $\Gamma_d = 9.8$ K/km is the dry adiabatic lapse rate, and $\Gamma_m$ is the moist adiabatic lapse rate at $T$ and $P$. In this paper, we set $z_0 = 500$ m, following Ragno and Hobbes (2005) and Wood (2009).

The optical depth of a liquid cloud with cloud depth H is given by Eq. (3):

$$\tau = \frac{3Q_{ext}}{4\rho_l}\int_0^H \frac{l}{r_e}\,dh.\tag{3}$$

$Q_{ext}$ is the extinction efficiency, $\rho_l$ is the density of water, and the effective radius is defined by

$$r_e = \frac{\int r^3 n(r)\,dr}{\int r^2 n(r)\,dr},\tag{4}$$

where $n(r)$ is the number concentration of cloud droplets with radius $r$. As Schulte et al. (2023) showed, when considering the assumptions of the subadiabatic model and the inherent relationship between $l$ and $n(r)$, it is possible to use Eqs. (1) and (3) to solve for $H$ and the profile of $l(h)$ given cloud optical depth and cloud top effective radius; that is, $\tau$ and $r_e(H)$. In practice, this is done using look-up tables because the analytical solution involves integrals which have no closed-form

expression. We refer the reader to Schulte et al. (2023) for details. Nevertheless, by using this method, we are able to convert estimates of $\tau$ and cloud top $r_e$, either from MODIS or from our random forest retrieval, into a modelled profile of cloud liquid



water. While not the focus of this paper, the subadiabatic model also produces an estimate of the total cloud droplet number concentration (N), which is assumed to be constant with height. Finally, we note that in order to provide for an apples-to-apples comparison against CloudSat products in Figs. 6 and 7, we average the subadiabatic profiles of LWC to the vertical

resolution of CPR. Specifically, we use a Gaussian-weighted moving average filter with a 6 dB window size of 480 m, and sample the filtered profile at the center of each CPR bin (every 240 m).

**2.3 Random Forest Regression Model**

Machine learning (ML), in general, refers to any empirical method whereby parameters are fit on a training dataset in order to optimize a predefined loss function (Chase et al., 2022). ML methods are based on statistical relationships between variables

rather than explicit physical models. Simple ML methods, such as linear regression, have been used in satellite retrievals for decades (e.g., Adler and Negri, 1988). More recently, more sophisticated methods which are better able to handle nonlinear relationships between variables have become more common (Hilburn et al., 2020; Hu et al., 2021; Yang et al., 2021; Zhang et al., 2021; Lee et al., 2022; Pfreundschuh et al., 2022; Goldenstern and Kummerow, 2023).

The random forest ML method, which we use in this study, is based on the concept of a decision tree (Breiman, 1984).

A decision tree is a hierarchical flowchart-like structure made up of decision nodes, branches, and leaf nodes. Each decision node represents a test that is performed on the input data (for example, whether the CPR surface return is above or below 10 dB), with the branches representing different possible outcomes of that test. The branch may lead to another decision node ("Is the wind speed above 5 knots?") or terminate in a leaf. At the leaf node, the tree provides the final model prediction. Decision trees can be used for either classification or regression problems, although our focus here is on regression. As the depth of a

decision tree increases, it often becomes over-fit to the training data (Chase et al., 2022). The random forest method (Breiman, 2001) attempts to compensate for this by using an ensemble of decision trees. Many different decision trees are created, each based on a random subset of training data sampled from the original dataset with replacement. When making a prediction, the random forest averages the results of all the decision trees in the ensemble. Recently, random forests have been used in atmospheric science to forecast severe weather (Hill et al., 2020), improve radar-based precipitation nowcasts (Mao and

Sorteberg, 2020), estimate particulate matter concentrations from satellite observations (Yang et al., 2021), and detect clouds (Haynes et al., 2022), among many other applications.

Our random forest model has 9 inputs and 2 outputs. The outputs are cloud optical depth and cloud top effective radius, and the model is trained to minimize the sum of the squared error between these predicted quantities and the corresponding MODIS observations. For training, we only use observations between 45° S and 45° N. This is because there

are biases in the MODIS cloud retrievals at high solar zenith angles (Grosvenor and Wood, 2014; Lebsock and Su, 2014), and we do not want the random forest to learn these biased relationships. Extrapolated retrievals can still be performed at these higher latitudes, however, just as they can be performed at night. The inputs are $TB_{94}$ and $\sigma_0$ from CPR; SST, TCWV, and 10 m wind speed from ECMWF-AUX; and CIAB, ODCOD $\tau$, CTER, and CTLWC from CALIOP. These inputs can be found in Table 1, along with our physical justification for including each of them. Several other input variables (for example, CloudSat-



derived path integrated attenuation) were tested; however, they were found to not significantly improve retrieval performance beyond what can be achieved with these 9 variables. It is also worth mentioning that two of the input variables, ODCOD $\tau$ and CTER, are CALIOP-based estimates of exactly the things we are trying to retrieve; that is, optical depth and cloud top effective radius. Essentially, the random forest takes the CALIOP-based estimates and adjusts them up or down depending on the additional information available in the other 8 inputs. This results in slightly better performance than just taking the CTER

estimates as-is, and much better performance than taking the ODCOD estimates as-is (because that product saturates so quickly). The model is trained using the Python package "scikit-learn" (Pedregosa et al., 2011). We include 100 trees in our forest, and each tree has a maximum depth of 50, with at least 50 samples required to be a leaf node. Other hyperparameters follow the default values in scikit-learn. The space of possible hyperparameter combinations one could choose is quite large and multidimensional; however, we performed a series of tests in which we retrained the model using stochastically chosen

combinations and found that the output of the model was not particularly sensitive to our choices.

| Input Parameter | Source (Instrument or Model) | Physical Justification |
|---|---|---|
| 94 GHz Brightness Temperature (TB$_{94}$) | CloudSat | Cloud water absorbs and re-emits microwave radiation emitted from the radiometrically cool ocean, increasing TB |
| CPR Surface Return ($\sigma_0$) | CloudSat | The ocean surface is very reflective, but the signal is attenuated by cloud water in the atmospheric column |
| Total Column Water Vapor (TCWV) | ECMWF HRES | Water vapor increases TB$_{94}$ and decreases $\sigma_0$ |
| Sea Surface Temperature (SST) | ECMWF HRES | TB$_{94}$ increases with SST. SST slightly modulates $\sigma_0$. |
| 10-m Wind Speed | ECMWF HRES | Wind speed affects ocean reflectance/emissivity and thus $\sigma_0$ and TB$_{94}$ |
| 532 nm Column Integrated Attenuated Backscatter (CIAB) | CALIPSO | A thicker cloud will scatter more, although this effect saturates quickly |
| Ocean Column Derived Optical Depth (ODCOD $\tau$) | CALIPSO | Similar to $\sigma_0$ – the ocean surface is reflective, but cloud water attenuates the signal |
| Hu et al. (2021) Cloud Top Effective Radius (CTER) | CALIPSO | Uses the full CALIOP profile and a machine learning algorithm to estimate $r_e$, one of our desired outputs |
| Hu et al. (2021) Cloud Top Liquid Water Content (CTLWC) | CALIPSO | Clouds with a higher CTLWC will tend to have higher optical depths |

**Table 1: The nine inputs to our random forest regression model, along with their sources and physical justifications for inclusion.**





## 3 Results

We first evaluate how well our model performs for daytime clouds, for which MODIS "ground truth" validation data is
available. Figure 2 includes density plots showing how model predictions of $\tau$, cloud top $r_e$, and LWP (i.e.., the height-integrated LWC from the subadiabatic model) compare to MODIS estimates for the same CloudSat pixels. These plots include all 2009 oceanic pixels detected as cloudy by MODIS between 45° S and 45° N and diagnosed as single-layer liquid clouds by CALIOP. There is overall good agreement for all 3 parameters, with Pearson's correlation coefficients of 0.74, 0.74, and 0.78 for $\tau$, $r_e$, and LWP, respectively. In other words, a little over half of the variance in these cloud quantities can be explained by
our predictive model. That said, model errors can still be quite large for individual cases, and the model predictions of $r_e$ (and to some extent $\tau$) are biased high at low values and biased low at high values. Additional summary statistics can be found in Table 2.

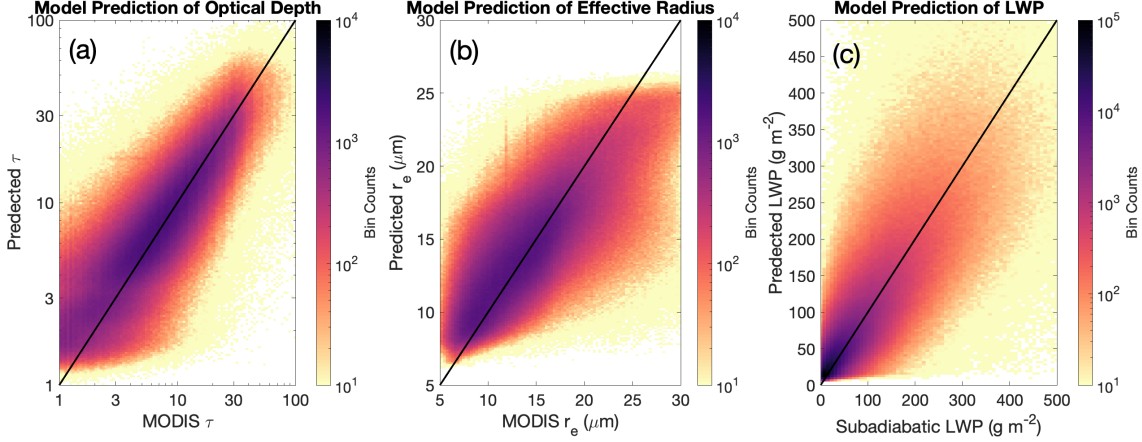

**Figure 2: Density plots showing model predictions of (a) cloud optical depth, (b) cloud top effective radius, and (c) cloud liquid water**
**path compared to MODIS retrievals, for daytime CloudSat oceanic pixels seen by MODIS in 2009 and identified as single layer liquid clouds by 2C-CLDCLASS-LIDAR. In the case of LWP, the MODIS-derived "Subadiabatic LWP" is calculated according to the method described in Schulte et al. (2023).**

| Parameter | Correlation Coef. | RMSE | MAE | Bias |
|---|---|---|---|---|
| Cloud optical depth ($\tau$) | 0.738 | 7.13 | 3.21 | +0.07 |
| Cloud top effective radius ($r_e$) | 0.735 | 3.38 $\mu m$ | 2.53 $\mu m$ | +0.06 $\mu m$ |
| Cloud liquid water path (LWP) | 0.779 | 63.5 g m$^{-2}$ | 30.0 g m$^{-2}$ | +1.23 g m$^{-2}$ |

**Table 2: Various model evaluation statistics for the year 2009, comparing the output of our random forest model to the MODIS**
**products the model is trained to emulate. RMSE is the root mean squared error, and MAE is the mean absolute error.**

With the random forest model, the latitude-weighted daytime oceanic warm liquid cloud fraction increases from 4.6% in 2B-CWC-RVOD to 23.5% (refer back to Fig. 1). There is a more modest increase in average warm cloud liquid water path from 6.4 g m$^{-2}$ in RVOD to 10.2 g m$^{-2}$ with the random forest model. This can be seen in Fig. 3, which plots maps of the





average unconditional daytime warm cloud LWP for the year 2009 from 2B-CWC-RO, 2B-CWC-RVOD, 1KM-AUX-
MODIS, and our random forest model. The pattern of average warm cloud LWP from the random forest model (bottom right panel) looks very similar to the MODIS map (bottom left), with slightly more (11% averaged over the globe) liquid water retrieved by the random forest model because it includes clouds which are detected by CALIOP but not by MODIS. Meanwhile, the average warm cloud LWP from 2B-CWC-RO is much higher than the other estimates, despite the cloud fraction being lower, which indicates that 2B-CWC-RO is likely retrieving cloud water contents that are too high for individual
clouds.

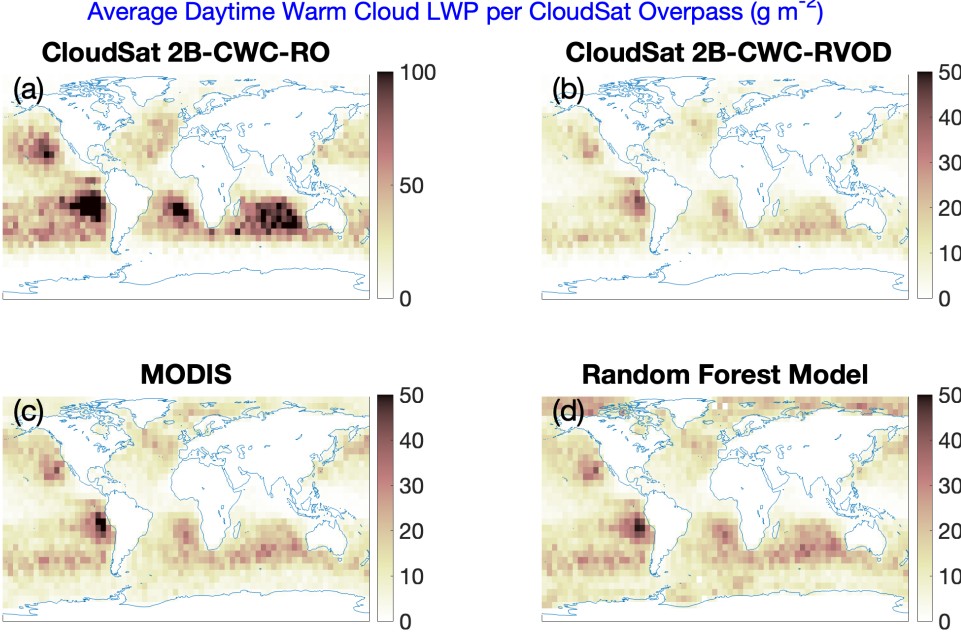

**Figure 3: Maps of 2009 average daytime cloud liquid water path over the oceans from (a) 2B-CWC-RO, (b) 2B-CWC-RVOD, (c) 1KM-AUX-MODIS, and (d) our random forest model. This is the unconditional average (the denominator is all daytime CloudSat overpasses). However, cloud liquid is only counted as part of the average (i.e. in the numerator) if a given pixel is identified as a**
**single layer liquid cloud by 2B-CLDCLASS-LIDAR. Note that panel (a) has a different color scale than the rest of the panels.**

Figure 4 plots histograms of model predictions of $\tau$, $r_e$, and LWP for 2009 daytime pixels, broken down into clouds that are thick enough to be detected by MODIS (in blue) and those that are missed by MODIS (in red). For clouds not seen by MODIS, the distributions of predicted $\tau$ and LWP heavily favor values near zero. For example, the median predicted LWP for these clouds is 17 g m$^{-2}$ compared to 46 g m$^{-2}$ for clouds that are seen by MODIS. This is encouraging to see, as we would
expect the clouds missed by MODIS to be thin and patchy, with low optical depths and liquid water paths.





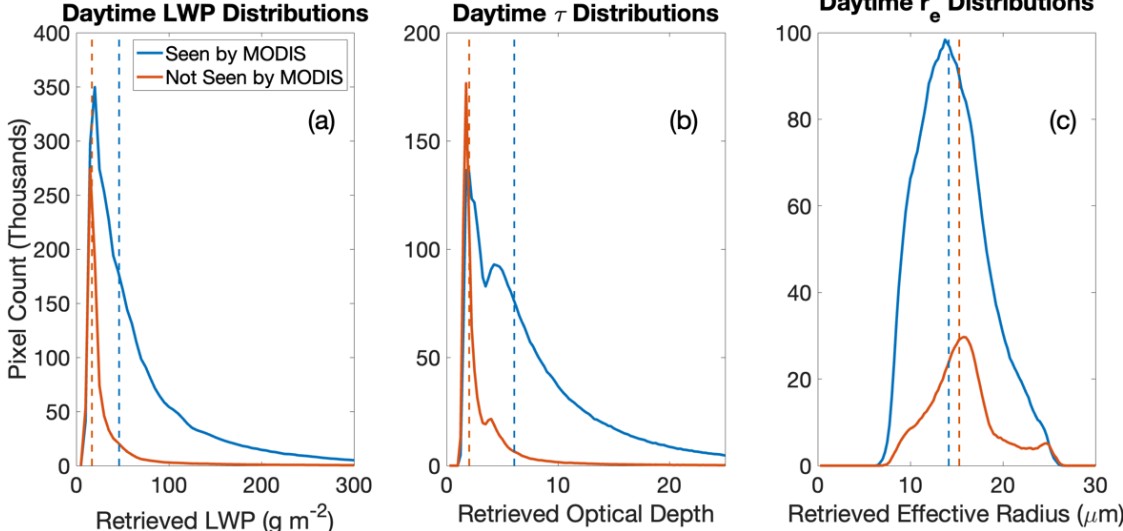

**Figure 4: Distributions of retrieved values of (a) liquid water path, (b) optical depth, and (c) cloud top effective radius from the random forest model for 2009 daytime pixels between 45° S and 45° N identified by 2B-CLDCLASS-LIDAR as single layer liquid clouds. The blue histograms correspond to pixels also identified as cloudy by 1KM-AUX-MODIS, and the red histograms to pixels identified by CALIPSO as cloudy but not by MODIS. The dotted vertical lines represent the median of each distribution.**

How does the retrieval algorithm perform at nighttime? While we lack nighttime observations with which to directly validate the retrievals (hence the need for a new algorithm in the first place), in Fig. 5 we compare the distributions of retrieved $\tau$, $r_e$, and LWP at night (in blue) to the distributions during the day (in red). The good performance of the model during the day, combined with the fact that the distributions in Fig. 5 are broadly similar, increases our confidence that the nighttime retrievals can be trusted. That said, there are some slight differences between the daytime and nighttime statistics. On average, the nighttime clouds (as retrieved) have slightly higher water paths and optical depths. While this finding is preliminary, and not the focus of this paper, it is consistent with previous studies (Wood et al., 2002; Burleyson et al., 2013; Giangrande et al., 2019) that have found higher LWPs at night in stratocumulus regimes. One proposed mechanism is that there is less turbulent coupling between the ocean surface and clouds during the day, depriving clouds of moisture and making them more susceptible to evaporation (Dong et al., 2014).

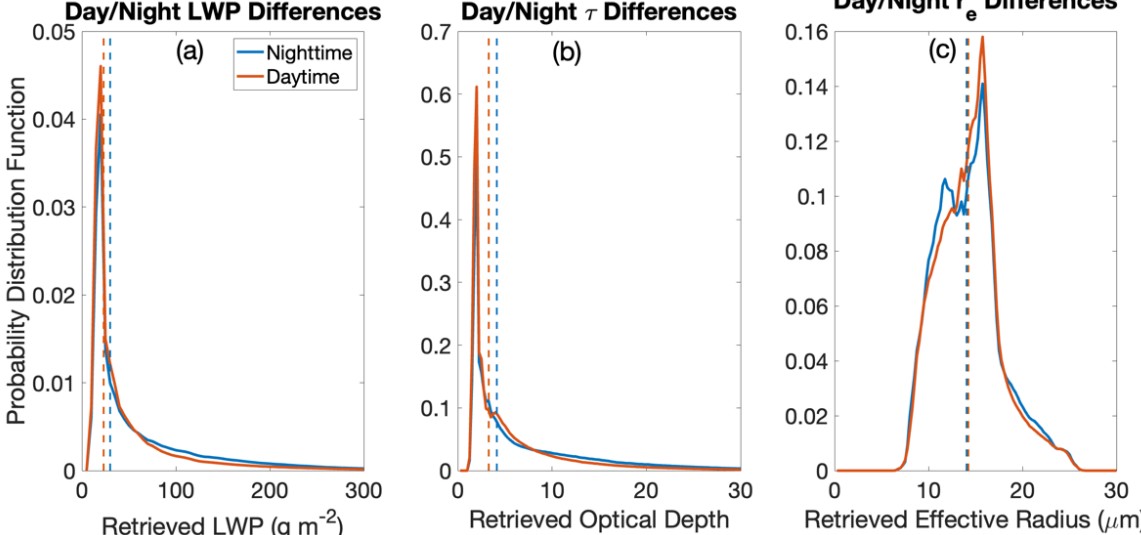

**Figure 5: Distributions of retrieved values of (a) liquid water path, (b) optical depth, and (c) cloud top effective radius from the random forest model for 2009 daytime (red) and nighttime (blue) pixels between 45° S and 45° N identified by 2B-CLDCLASS-LIDAR as single layer liquid clouds. The blue histograms correspond to pixels also identified as cloudy by 1KM-AUX-MODIS, and the red histograms to pixels identified by CALIPSO as cloudy but not by MODIS. The dotted vertical lines represent the median of each distribution.**

### 3.1 Case Studies

We now turn our attention to two case studies, which help demonstrate our algorithm's usefulness for estimating the liquid water content of thin liquid clouds. The case studies come from two randomly chosen 2009 CloudSat granules that included observations over the subtropical southeastern Pacific Ocean, an area with persistent stratocumulus cloud decks. Figure 6 includes several plots from the first case study, which occurred during the daytime on 2 September 2009. The left side panels show cross sections of various observed and retrieved variables along a portion of the CloudSat orbital track. The CALIOP 532 nm total attenuated backscatter (TAB; top left panel) indicates a cloud top that occurs at around 1.25 km in altitude along nearly the entirety of this ~500 km long cross section. From the CPR W-band reflectivity field, however, only portions of this cloud deck are distinguishable from the background noise. This means that about half of the cloudy pixels (according to CALIOP) have no LWC profile in the 2B-CWC-RVOD product. The next panel down shows the LWC profile along this cross section according to our random forest algorithm, once the subadiabatic model has been applied to our retrievals of $\tau$ and $r_e$, and the bottom left panel shows a "merged" LWC cross section that uses the 2B-CWC-RVOD LWC profile where it is non-zero, but fills in the gaps with the random forest result for pixels where 2B-CWC-RVOD does not detect a cloud. For the clouds that are thick enough to be seen by CloudSat, while the random forest predictions do not match the 2B-CWC-RVOD profiles exactly, there is general agreement as to the depth of the cloud, the order of magnitude of LWC values, and as to which pixels have the highest LWC values. In fact, there is excellent agreement between the two retrievals when it comes to the integrated liquid water path for these CloudSat-detected pixels, as demonstrated in the bottom right panel. The aim of our





product, however, is not to replace the reflectivity-based retrieval but to supplement it in the cases where the radar does not

detect a cloud. To this end, it is encouraging that the merged LWC cross section looks quite reasonable, without any sharp

discontinuities. Also included in Fig. 6 are the predicted $\tau$ and $r_e$ values for this cross section from the random forest model

compared to MODIS and (in the case of $r_e$) compared to 2B-CWC-RVOD. For this case, the retrieved optical depth tracks

almost exactly with MODIS, while the effective radius also generally follows the MODIS line but not quite as closely. Finally,

Fig. 6 plots 3 of the most important inputs to the random forest model: TB$_{94}$, $\sigma_0$, and CIAB. Where the clouds are thickest,

TB$_{94}$ is higher, $\sigma_0$ is lower, and CIAB tends to be higher (although that measurement is noisier).

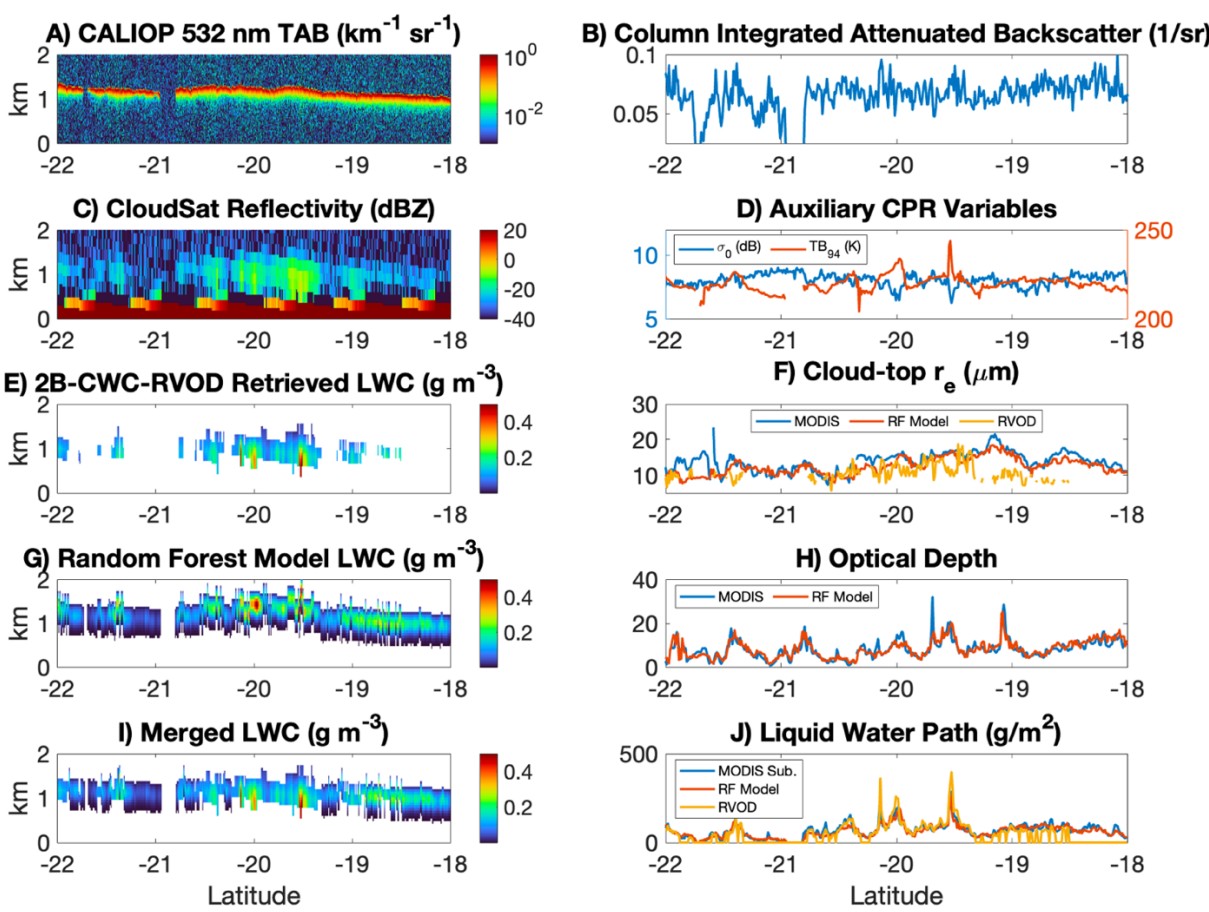

**Figure 6: Daytime case study from 2 September 2009 (CloudSat granule 17816). A) CALIOP total attenuated backscatter; B) CALIOP column integrated attenuated backscatter; C) CPR reflectivity; D) CPR surface return and 94 GHz brightness temperature; E) Cloud liquid water content profiles retrieved by 2B-CWC-RVOD algorithm; F) Cloud-top effective radius as**
**estimated by MODIS (blue), our random forest model (red), and 2B-CWC-RVOD (gold); G) Profiles of LWC retrieved by applying the subadiabatic model to the random forest retried values of $\tau$ and $r_e$; H) Optical depth from MODIS and the random forest model; I) Merged profile of LWC, which supplements the 2B-CWC-RVOD retrieval with random forest profiles for cloudy pixels that have no 2B-CWC-RVOD retrieval; J) Vertically integrated cloud liquid water path from MODIS (using the subadiabatic model), the random forest algorithm, and 2B-CWC-RVOD.**






The second case study (Fig. 7) is a nighttime case from 30 June 2009. Once again, CALIOP indicates a much less broken cloud deck than the CloudSat LWC retrieval (2B-CWC-RO in this case, since 2B-CWC-RVOD is daytime-only). The random forest LWC profiles are not quite as deep from cloud base to cloud top as 2B-CWC-RO, leading to higher maximum LWC values in the random forest output. Still, the merged LWC cross section looks decent, and certainly closer to reality than

2B-CWC-RO alone. There is also reasonable agreement between the LWP retrieved by 2B-CWC-RO and the random forest, although they disagree on the cloud top effective radius. We see a similar pattern in the inputs as is present for the daytime case study: higher LWP is associated with higher $TB_{94}$ and CIAB, and lower $\sigma_0$, although the pattern is noisier for this case than the daytime case.

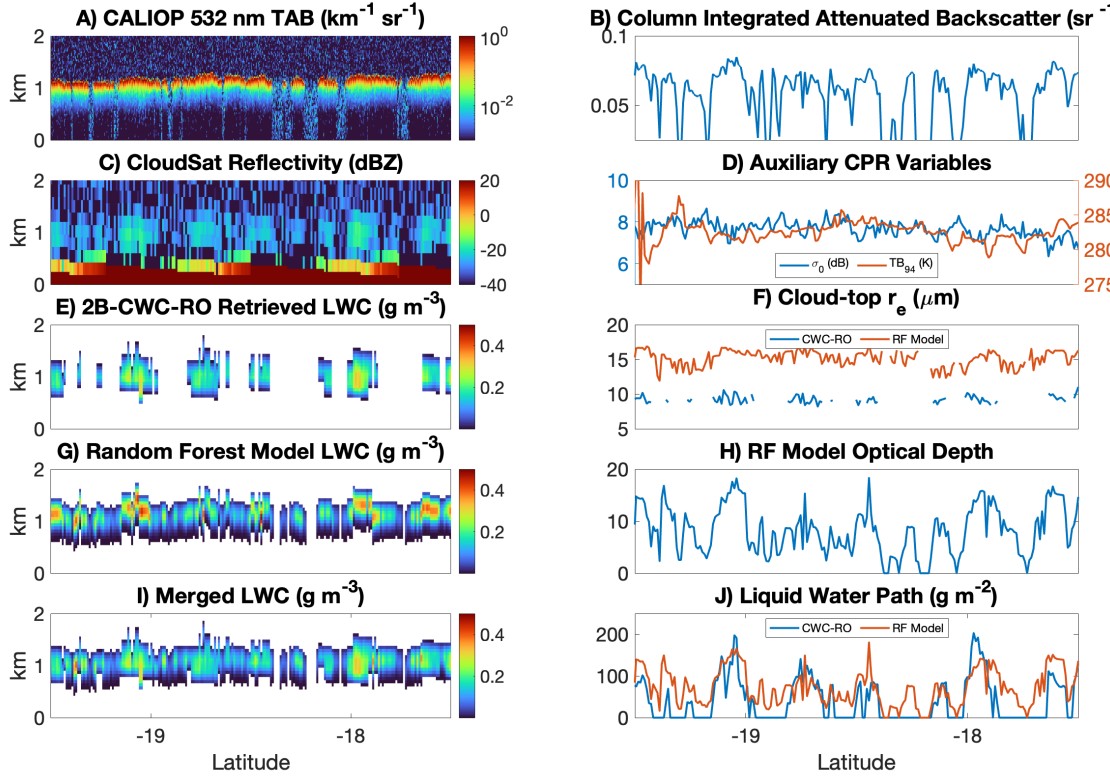

**Figure 7: Nighttime case study from 30 June 2009 (CloudSat granule 16877). As for Fig. 6, except with 2B-CWC-RO replacing 2B-CWC-RVOD (which is unavailable at night) in panel E.**



## 3.2 Input variable importance

We now turn our attention to the question of how our random forest model reaches the predictions that it does. Each of the
inputs of the model was chosen because we had reason to believe there would be a physical relationship between the input
variable and one of the target variables ($\tau$ or $r_e$). For example, the ocean has a relatively low emissivity at 94 GHz, meaning
that clear sky pixels will appear colder than those with clouds. On the other hand, cloud water attenuates the radar pulses, so
greater cloud water will lead to a reduced radar return from the ocean surface, all other things equal (Lebsock et al., 2022).
And of course, since they are attempting to measure the same thing, it is to be expected that CTER from CALIOP should be
related to MODIS cloud top $r_e$. Figure 8 demonstrates these relationships with density plots. These relationships become even
stronger once accounting for confounding environmental variables. Water vapor also absorbs microwave radiation at 94 GHz,
so higher TCWV will increase TB$_{94}$ and decrease $\sigma_0$, much in the same way as a cloud. A higher SST will increase surface
emission and will thus also increase TB$_{94}$. And wind speeds affect the backscatter characteristics of the ocean surface, with
lower wind speeds tending to lead to higher but also much more variable surface returns.

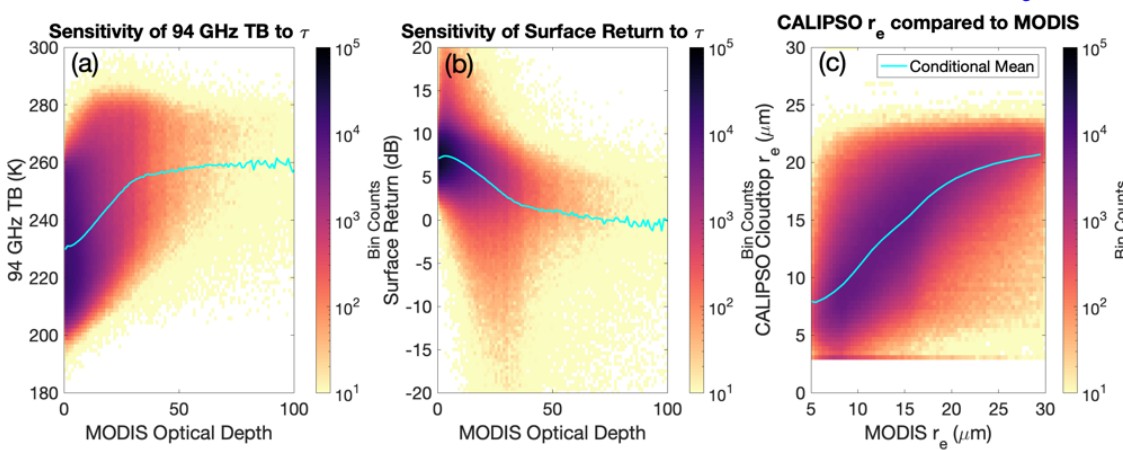

**Figure 8: Density plots showing (a) CloudSat 94 GHz brightness temperature, (b) CloudSat surface return, and (c) CALIPSO cloud top $r_e$ from Hu et al. (2021), each compared to corresponding MODIS observations from 2008. In each panel the cyan line shows the conditional mean of the variable on the y axis, conditioned on the variable on the x-axis.**

Table 3 shows the partial correlation (e.g., Baba et al., 2004) between each input variable and MODIS $\tau$, $r_e$, and LWP,
accounting for TCWV, SST and wind speed. By this metric, the most important variables for predicting $\tau$ and LWP are TB$_{94}$,
$\sigma_0$, and CIAB, while the most important for predicting $r_e$ is (unsurprisingly) CTER. A simpler model than our random forest
can be constructed that exploits these linear relationships. We fit a multinomial linear regression model to our training dataset
of 2008, using the same 9 variables as the random forest model, and then tested the regression model on the 2009 data, and we
got correlation coefficients of 0.62 and 0.71, respectively, for $\tau$ and $r_e$. These are decent correlation coefficients, even if they
are smaller than those obtained from the random forest model.



| Variable | Optical Depth Corr. | Cloud top $r_e$ Corr. | LWP Corr. |
|---|---|---|---|
| TB94 | 0.625 | 0.410 | 0.711 |
| $\sigma_0$ | -0.493 | -0.308 | -0.603 |
| CIAB | 0.448 | -0.056 | 0.376 |
| ODCOD $\tau$ | 0.373 | 0.047 | 0.349 |
| CTLWC | 0.243 | 0.046 | 0.220 |
| CTER | 0.042 | 0.665 | 0.221 |

**Table 3: Partial correlation coefficients (controlling for the environmental variables of TCWV, SST, and 10 m wind speed) between the various CloudSat and CALIPSO input variables and the MODIS target variables of cloud optical depth, effective radius, and liquid water path. The data comes from the 2008 training dataset.**

Why is the random forest model able to do better than the multilinear regression model? We speculate this is because it can interpret nonlinear relationships in the data. For example, several of the input variables saturate at high optical depths. This is most extreme for the ODCOD $\tau$ variable, which saturates at cloud optical depths of about 3 during the daytime and 5 during the nighttime. As evidenced in Fig. 8, though, the TB94 signal saturates at a cloud optical depth around 30, and for $\sigma_0$ saturation is reached closer to an optical depth of 50. Another example of a nonlinear relationship is the fact that the $\sigma_0$ variable is more predictive at higher wind speeds than at lower wind speeds. The linear correlation coefficient between $\sigma_0$ and MODIS $\tau$ is -0.43 for pixels with wind speeds between 7 and 10 m/s, while it is only -0.24 for pixels with wind speeds between 0 and 3 m/s.

To test which variables are most important to the random forest model, specifically, we use a method called backwards sequential feature selection (Aha and Bankert, 1996). Starting with the full list of 9 input variables, we train 9 different random forest models to predict MODIS $\tau$, each missing exactly one of the 9 input variables. For computational reasons, we do not use the full 2008 training dataset but only a subset consisting of a random 5 %. Each of the resulting candidate models is evaluated against the test dataset, and we search for the model which has the highest correlation between predicted $\tau$ and MODIS $\tau$. The variable that is missing from this best model is deemed the least important variable for predicting $\tau$ (in this case, that variable is CTLWC). Then we repeat the process with the remaining 8 variables. We train 8 new models, each missing exactly one of the remaining variables, and search for the model that performs best. This process is iteratively repeated until only one variable is left. Similarly, we use backwards sequential feature selection to determine the most important variables for predicting MODIS $r_e$. The results are plotted in Fig. 9. According to this method, the single most important variable for predicting MODIS $\tau$ is TB94, while the most important variable for predicting MODIS $r_e$ is CTER (by far). TCWV also ranks highly in both lists, probably because knowing the amount of water vapor greatly improves the utility of the TB94 measurement. Note that this method does not explicitly account for the correlations among the different input variables, which influences the features identified as most important. For example, if TB94 were unavailable, one would expect $\sigma_0$ to be most important for predicting $\tau$. Because TB94 and $\sigma_0$ are not independent of each other, $\sigma_0$ ranks as less important according to the backwards sequential feature selection algorithm.

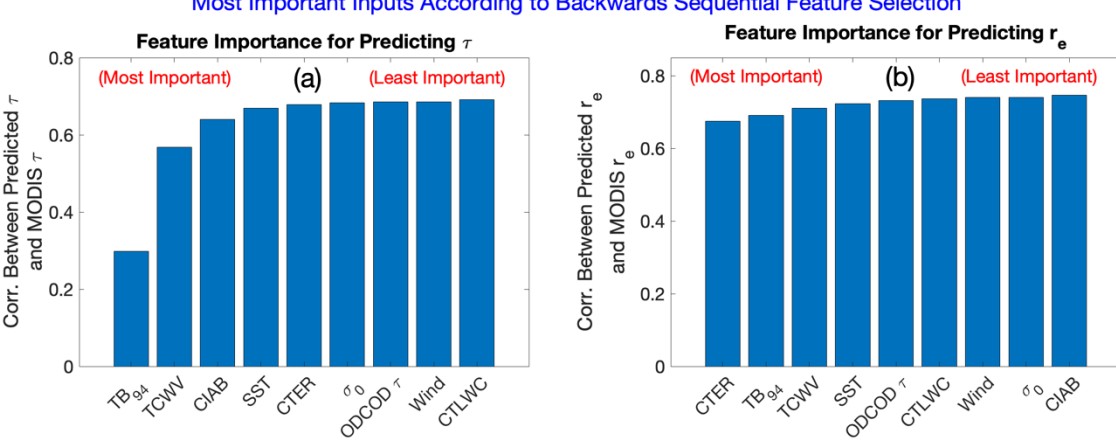

**Figure 9: (a) Each bar shows the correlation between predicted optical depth and MODIS optical depth, for a model trained and**
**tested using only the feature below each bar, plus all variables to the left. The variables are listed from left to right according to their importance rank using the backward sequential feature selection algorithm. Note that adding the first few features greatly improves model performance, but that there are diminishing returns to adding additional features. (b) As for panel (a), but for cloud top effective radius.**

## 4 Conclusions

Many thin liquid clouds do not produce W-band reflectivities above the CloudSat radar noise floor or the surface clutter noise. The current operational cloud water content retrieval products thus do not include these clouds, which complicates radiative flux calculations and makes comparisons to climate models more challenging. However, even if these clouds do not show up in CPR reflectivities, there is a significant amount of information about the clouds in other CloudSat observables (in particular, $TB_{94}$ and $\sigma_0$), and in the near-coincident measurements available from CALIPSO. It is this information that we aim to leverage

using our random forest model. While machine learning based models are often thought of as "black boxes," we select input variables that we expect will be related to the cloud properties of optical depth and cloud top $r_e$ through clearly-defined physical mechanisms. Making additional assumptions (i.e., those of the subadiabatic model), it is straightforward to derive estimated profiles of cloud water.

While the resulting LWC profiles certainly have flaws, and should not be expected to perfectly capture the vertical

distribution of cloud water, there is great potential for them to be used to augment reflectivity-based estimates of liquid cloud water, filling in the gaps in cases where we know (from CALIOP and/or MODIS) that a cloud is present, but not detected by CPR. The effects of including these thin clouds are large. With the random forest model, the daytime oceanic warm cloud liquid cloud fraction increases about five-fold compared to 2B-CWC-RVOD, while the total warm cloud LWP amount nearly doubles. The model gives comparable results to the MODIS-based method presented in Schulte et al. (2023); however, this

method does not use observations that rely on reflected sunlight, so it can be used during the night.

The method is not without limitations. Many of the input variables are only useful over the ocean, and we have not considered mixed-phase or multi-layered clouds. It is also worth noting that CPR, CALIOP, and MODIS observations are not

*perfectly* coincident, and that they have different resolutions. The assumptions of the subadiabatic model should not be expected to hold true in all cases, and both this study and Schulte et al. (2023) suggest that the subadiabatic model might

generate clouds that are too vertically compressed (with a cloud base that is too high) for pixels with high optical depths. Still, the case studies that we have shown demonstrate that when one merges the random forest LWC estimates with profiles from 2B-CWC-RVOD or 2B-CWC-RO, generally realistic-looking curtains of LWC are obtained.

We intend to include random forest predictions of oceanic cloud properties (including $\tau$, $r_e$, LWP, and cloud droplet number concentration) and LWC profiles in the final reprocessed version of the 2B-CWC-RVOD product. The method could

also easily be extended to future satellite missions. The EarthCARE mission (Illingworth et al., 2015), set to launch in May 2024, will include both a 94 GHz radar as well as a 355 nm lidar and MODIS-like instruments. While retraining would be necessary due to instrument differences, our random forest method could be used to supplement EarthCARE LWC profile estimates for thin clouds. A lidar and cloud-sensitive radar are also being planned for the polar orbiting satellite of NASA's Atmosphere Observing System (AOS). This radar is likely to be even less sensitive to thin clouds than CPR, meaning non-

reflectivity based strategies of estimating liquid cloud properties will be all the more important.

**Code availability**

All code used to produce the results presented in this study is available from the Zenodo repository (https://doi.org/10.5281/zenodo.10425919).

**Data availability**

All of the CloudSat and MODIS data used in this study, along with the CALIOP estimates of CTER and CTLWC, are available from the CloudSat data processing center at cloudsat.cira.colostate.edu (last access: 22 Dec 2023). The remaining CALIOP data is available from the NASA Atmospheric Science Data Center at asdc.larc.nasa.gov/project/CALIPSO (last access: 22 Dec 2023). Other data necessary to reproduce the presented results are available on request.

**Author contributions**

RS performed the data analysis and wrote most of the article. ML, JH, and YH helped conceptualize and focus the study, provided technical help and discussions, and helped edit the article.

**Competing interests**

The authors declare that they have no conflict of interest.



**Acknowledgments**

This work was funded by the National Aeronautics and Space Administration's *CloudSat* mission. The work of ML was performed at the Jet Propulsion Laboratory, California Institute of Technology, under a contract with NASA.

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
