# Peer review of "A random forest algorithm for the prediction of cloud liquid water content from combined CloudSat/CALIPSO observations"

_Atmospheric Measurement Techniques, 2023_

## Referee Comment (RC2)

**Review of:**

**„A random forest algorithm for the prediction of cloud liquid water content from combined CloudSat/CALIPSO observations"**

**by Richard Schulte et al.**

**General comment**

In the manuscript named „A random forest algorithm for the prediction of cloud liquid water content from combined CloudSat/CALIPSO observations", the authors use machine learning (random forests) to predict cloud optical depth and cloud top effective radius with CloudSat and CALIPSO observables, independent of the typically used radar reflectivity profile. The idea is to a) fill gaps in existing CloudSat radar-based products during daytime and b) estimating cloud water profiles during nighttime where this information is missing completely.

The manuscript touches on an important topic for climate/cloud research, in particular the potential of the described method to derive nighttime cloud microphysics is large, and of interest to the readers of AMT. The manuscript is well written and the figures support the conclusions drawn by the authors. I recommend the manuscript to be published after my minor concerns and comments are adequately addressed.

**Minor points that need revision**

- I was a bit surprised by the authors choice regarding the architecture of the random forests: with about 25 million data points during training, the authors train only 100 trees, but balance this by allowing to grow the trees very deep (max depth of 50). This is somewhat opposing the original idea of random forests: to have many weak (shallow) decorrelated learners. I would assume that with the current model choice the model setup it would overfit, but this is not reported.

- The authors evaluate the random forest predictions with MODIS observations, and aim at using the data for filling the gaps of the CloudSat 2b-CWC-RVOD LWC data. In my opinion, it would be good to provide an additional evaluation for these specific situations (during daytime), as a) the situations where gaps exist tend to feature particularly low LWCs (thus insufficient reflectivity) and b) the random forests tend to overestimate cloud optical thickness and effective radius when their values are low (i.e. in these situations, as shown in figure 2). This also somewhat affects the use case of filling the gaps, as these gaps are likely filled with this bias.

- The authors analyze the importance of the random forests input features. I appreciate this, but it does not really help to provide an answer for the speculation that the improved skill of the random forests (compared to the linear regression) is due to the nonlinear capabilities (on page 15), e.g. capturing the saturation of the TB94 signal. This could be easily analyzed using partial dependencies.

- Random forests are incapable of extrapolating, and I am wondering if 1 year of data is enough to capture all variability of the input feature distributions (the authors argue that there is a „slightly different climate" between 2008 (training) and 2009 (test data)). I am interested to learn if the authors have compared the distributions of the input data between the training and test data. If there are differences in the distributions, a different machine learning technique would likely be more appropriate.

---

## Author Response (AR1)

**Reviewer #1 Comments**

**General Comments**
The authors make a compelling case that their algorithm achieves additional coverage of cloud water content relative to the standard CloudSat radar retrieval algorithms (2B-CWC-RVOD and 2B-CWC-RO) in cases where the clouds are too thin to be detected by the radar. However, I wonder what fraction of the additional coverage corresponds to truly optically thin clouds that cover the entire horizontal pixel and what fraction corresponds to clouds that partially cover the pixel and therefore appear thin. The first case could lead to high quality retrievals of cloud water content because it satisfies the assumption of a horizontally uniform cloud used in the sub-adiabatic cloud model, but the second case may not. It would strengthen the paper if the authors could perform a quality control analysis that estimates the frequency of occurrence of these two cases. This could be done with the MODIS flag for partly cloudy pixels, or perhaps with some other information from CloudSat/CALIPSO.

This is a good question. The table below (Table 3 in the new manuscript) breaks down all daytime pixels identified as cloudy by CALIPSO into 6 categories, based on whether or not they are visible to the CloudSat radar and how they are classified by MODIS. The largest category (category 2) is pixels that are identified as fully cloudy by MODIS but aren't detected at all by the CloudSat radar. For these pixels, the random forest retrieved liquid water paths agree quite well with the MODIS retrievals. It is true that there are a large number of pixels for which CALIOP indicates a cloud but MODIS either does not recognize a cloudy or identifies the pixel as partly cloudy. We have added discussion in the paper (lines 227-232) about how the sub-adiabatic model may be deficient in these cases.

| Category | % of all pixels identified as cloudy by CALIOP | Median RF retrieved LWP | Median MODIS subadiabatic LWP | Median RVOD retrieved LWP | LWP correlation between retrieval and MODIS | LWP correlation between retrieval and RVOD |
|---|---|---|---|---|---|---|
| 1: MODIS cloudy / radar detected | 18.2 | 88.2 | 90.0 | 101.2 | 0.71 | 0.67 |
| 2: MODIS cloudy / radar undetected | 31.3 | 34.2 | 30.3 | - | 0.81 | - |
| 3: MODIS PCL / radar detected | 2.4 | 35.6 | 29.8 | - | 0.38 | - |
| 4: MODIS PCL / radar undetected | 18.4 | 16.8 | 11.0 | - | 0.46 | - |
| 5: MODIS undetected / radar detected | 2.5 | 38.2 | - | - | - | - |
| 6: MODIS undetected / radar undetected | 27.2 | 15.6 | - | - | - | - |

The case studies in Section 3.1 demonstrate that the new coverage attainted by the author's retrieval algorithm can complement the data from the existing CloudSat operational retrievals without any clear discontinuities or artifacts. This shows that the new retrieval algorithm can add

value to the existing retrieval algorithms when the new algorithm detects a cloud, but the existing algorithms do not. However, I think it would improve the paper if the authors could also do a statistical comparison of the cloud water content in the range bins in which both the new algorithm and the existing algorithm detect a cloud. This would show a more complete evaluation of how well the new data fits with and complements the existing data.

The figure below shows how the retrieved cloud liquid water path, column maximum liquid water content, and cloud top effective radius from the random forest algorithm compares to the retrieved values from 2B-CWC-RVOD, for pixels where RVOD detects a cloud. In general, there is excellent agreement on the integrated amount of cloud water, but the random forest model tends to predict clouds that are slightly more condensed in the vertical dimension that what RVOD detects, with correspondingly higher column max LWCs and cloud top effective radii. The distribution of cloud top effective radius also has less spread in the random forest results than we see in RVOD (though, in the case of RVOD, there could be some contamination from drizzle drops). These results are consistent with what we found in Schule et al. (2023), when we compared the MODIS subadiabatic model with the RVOD retrieval.

[Figure]

Because we did an in-depth comparison between the MODIS subadiabatic model and RVOD in the previous paper, we chose to focus on comparing our new results against MODIS observations and the MODIS subadiabatic model. Our hope is that by showing that this new algorithm agrees well with MODIS, we will convince the reader that it also agrees well with 2B-CWC-RVOD, while keeping the paper to a reasonable length. Nevertheless, we agree that this is an important question that is now addressed more completely in the paper, including in the new Table 3 and in the discussion in lines 185-187, 222-224, and 386-389.

**Specific Comments**

Fig. 1: panel (b) has a different latitude range than the other panels. Consider changing this so that all panels have the same latitude range to make it easier to compare.

Thank you for pointing out this oversight. It has been corrected in the latest version of the manuscript.

Section 2.2: This section clearly describes the method for estimating the vertical profile of cloud liquid water content relative to the height above cloud base, l(z), and the cloud geometric thickness, H. However, I believe that the height of the cloud base also needs to be known in order to estimate the cloud water content profile as a function of the height above sea level. I could not find the explanation of how the cloud-base height is estimated. Can you please explain this?

As you say, this section describes how we solve for the cloud geometric thickness, H, and the vertical profile of LWC relative to cloud base, which we call l(h) in the paper. To determine the cloud base, and therefore l(z) relative to sea level, we set the top of the cloud according to the CALIOP detected cloud top height using the 2B-CLDCLASS-LIDAR product (this is now emphasized in lines 133-134). The cloud-base height is then $z_{top}$ – H. In the extremely rare cases where our estimate of H is larger than $z_{top}$ (i.e., where the subadiabatic model predicts a cloud that extends below sea level), we iteratively increase the assumed adiabatic condensation rate until H < $z_{top}$. These procedures are detailed in Schulte et al. (2023).

Fig. 2: It is difficult to distinguish colors between 10^3 and 10^5 counts. Consider adding contours of counts to improve the clarity of the figure.

Thank you for bringing this to our attention. Contour lines have been added to the figure.

Section 3.1 Case studies: Throughout the paragraph starting on line 243, it would help to refer the panel labels in Fig. 6 (e.g. Fig. 6a, Fig. 6b, etc.). This would be clearer than the wording "the next panel down" etc., which is currently used in the text.

Thank you for this good suggestion. It has been incorporated into the new draft.

**Reviewer #2 Comments**

I was a bit surprised by the authors choice regarding the architecture of the random forests: with about 25 million data points during training, the authors train only 100 trees, but balance this by allowing to grow the trees very deep (max depth of 50). This is somewhat opposing the original idea of random forests: to have many weak (shallow) decorrelated learners. I would assume that with the current model choice the model setup it would overfit, but this is not reported.

Thank you for this observation. While we did test models using a larger number of trees, no significant improvement was found past 100 trees (if using a max tree depth of 50). As evidence of this, the table below shows the test dataset correlation in optical depths between the random forest model and MODIS for models with either 100 or 1000 trees and with max depths of either 10 or 50. From this it appears that one can obtain similar results using either a plethora of shallower trees or a modest number of deeper trees. We have chosen a smaller number of trees in this paper for computational expediency, though we will consider other choices for the operational algorithm. In any case, for the configuration we use in the paper (100 trees; max depth 50), there is no evidence that the model is overfit. Performance on the training dataset is comparable to the performance on the test dataset, and we do not get better results when increasing the number of trees or decreasing their max depth. The new manuscript notes these results in lines 179-180 and 194-196.

|  | 100 Trees | 1000 Trees |
|---|---|---|
| Max tree depth = 10 | 0.706 | 0.739 |
| Max tree depth = 50 | 0.738 | 0.739 |

The authors evaluate the random forest predictions with MODIS observations, and aim at using the data for filling the gaps of the CloudSat 2b-CWC-RVOD LWC data. In my opinion, it would be good to provide an additional evaluation for these specific situations (during daytime), as a) the situations where gaps exist tend to feature particularly low LWCs (thus insufficient reflectivity) and b) the random forests tend to overestimate cloud optical thickness and effective radius when their values are low (i.e. in these situations, as shown in figure 2). This also somewhat affects the use case of filling the gaps, as these gaps are likely filled with this bias.

This is a good point. For daytime clouds missed by 2B-CWC-RVOD (because they are missed by CPR) but seen by MODIS (i.e., category 2 in Table 3), we are confident in the quality of the random forest predictions. For clouds missed by both CPR *and* MODIS, there is a greater possibility of bias. Still, it is very hard to evaluate the model predictions in these instances because there is little to compare against. The one comparison we can make is to compare random forest predictions of cloud optical depth against ODCOD predictions of optical depth, for pixels that are identified as cloudy by CALIOP but not by MODIS. Note, however, that this is not a perfect comparison because many of the pixels in this category are classified as either missing or saturated by ODCOD. In addition, the ODCOD optical depth is a total optical depth that includes aerosol contributions. Still, the figure below shows that higher optical depths from

the random forest are associated with higher ODCOD optical depths, but that random forest estimates are biased high by maybe half an optical depth or so.

We have added a paragraph in the paper about these issues (lines 219-232), emphasizing the potential for bias in filling the gaps for the thinnest clouds. It is worth noting that, during the night, no 2B-CWC-RVOD estimates are available for even the thickest of clouds, and so this bias will be much less of a concern. We believe that slightly biased predictions of liquid water content are better than having no predictions at all.

[Figure]

Note: The red line above shows the median random forest value in each bin. The ODCOD signal saturates at an optical depth of around 2 during the daytime. These saturated pixels are excluded.

The authors analyze the importance of the random forests input features. I appreciate this, but it does not really help to provide an answer for the speculation that the improved skill of the random forests (compared to the linear regression) is due to the nonlinear capabilities (on page 15), e.g. capturing the saturation of the TB94 signal. This could be easily analyzed using partial dependencies.

Thank you for the suggestion to use partial dependencies. The figure below shows the partial dependence of optical depth on two selected input variables, CPR surface return and CALIOP column integrated attenuated backscatter (the vertical black lines show the deciles of the distribution of each input variable). These two variables clearly show nonlinear relationships. We have eliminated the speculative wording and have added a comment about partial dependencies in lines 341-343.

Partial Dependence of MODIS Cloud Optical Depth

$\sigma_0 * 100$            CIAB

Random forests are incapable of extrapolating, and I am wondering if 1 year of data is enough to capture all variability of the input feature distributions (the authors argue that there is a „slightly different climate" between 2008 (training) and 2009 (test data)). I am interested to learn if the authors have compared the distributions of the input data between the training and test data. If there are differences in the distributions, a different machine learning technique would likely be more appropriate.

The plot below shows the distributions of all input variables for 2008 and 2009. The distributions are quite similar, so we are not concerned about the random forest being forced to extrapolate. To avoid confusing the reader, we have eliminated the sentence mentioning a "slightly different climate."

[Figure]

Distributions of Input Variables